# Plasma Biomarkers of Mitochondrial Dysfunction in Patients with Myasthenia Gravis

**DOI:** 10.3390/medsci13030118

**Published:** 2025-08-08

**Authors:** Elena E. Timechko, Marina I. Severina, Alexey M. Yakimov, Anastasia A. Vasilieva, Anastasia I. Paramonova, Natalya V. Isaeva, Semen V. Prokopenko, Diana V. Dmitrenko

**Affiliations:** 1Russian National Research, V.F. Voyno-Yasenetsky Krasnoyarsk State Medical University, 660022 Krasnoyarsk, Russianv_isaeva@mail.ru (N.V.I.);; 2Krasnoyarsk Regional Clinical Hospital, 3A, Partizana Zheleznyaka str., 660022 Krasnoyarsk, Russia; 3Federal Siberian Research Clinical Center, Russian Federal Medical Biological Agency, 660022 Krasnoyarsk, Russia

**Keywords:** myasthenia gravis, mitochondrial dysfunction, microRNA, biomarkers

## Abstract

**Background**. Myasthenia gravis is an autoimmune neuromuscular disease characterized by fatigue of striated muscles due to impaired neuromuscular transmission. Mitochondrial dysfunction, according to published data, contributes significantly to metabolic abnormalities, oxidative stress and, as a consequence, the persistence of inflammation. MicroRNAs, which are post-transcriptional regulators of expression, are able to contribute to the aberrant functioning of mitochondria. In this study, with the aim of searching for biomarkers at the level of circulating microRNAs and proteins, the expression of three microRNAs was analyzed and the concentration of mitochondrial proteins was measured in the blood plasma of patients with myasthenia gravis (*n* = 49) in comparison with healthy volunteers (*n* = 31). **Methods**. Expression analysis was performed by RT-PCR, mathematical data processing was carried out using the Livak method, and protein concentration was determined by enzyme immunoassay. **Results**. Our plasma expression analysis revealed a statistically significant increase in hsa-miR-194-5p expression (Log10 Fold Change = 1.46, *p*-value < 0.0001) and a statistically significant decrease in hsa-miR-148a-3p expression (Log10 Fold Change = −0.65, *p*-value = 0.02). A statistically significant decrease in plasma COQ10A concentration was also found (0.911 [0.439; 1.608] versus 1.815 [1.033; 2.916] for myasthenia gravis and controls, respectively, *p*-value = 0.01). **Conclusion**. Our data suggest hsa-miR-148a-3p and hsa-miR-194-5p, as well as COQ10A, as potential biomarkers of mitochondrial dysfunction in myasthenia gravis.

## 1. Introduction

Myasthenia gravis is an autoimmune disease characterized by chronic muscle fatigue and skeletal muscle weakness [1], with a prevalence of 40–180 people per million [2]. Clinical manifestations vary depending on the type of autoantibodies involved, as well as the presence or absence of thymic neoplasms [3]. The main cellular mediators are T cells and B cells [4]. The pathological process is induced by the production of autoantibodies, in most cases against the nicotinic acetylcholine receptor (AChR)—recorded in about 85% of patients [5]—and against muscle-specific tyrosine kinase (MuSK)—recorded in about 7% of patients [6]. Also, in some cases, autoantibodies to the low-density lipoprotein receptor-associated protein 4 (Lrp4) were recorded [7].

The diagnosis is based on the detection of the above-described autoantibodies in the blood serum [8]. However, for some cases, the so-called seronegative form of myasthenia is registered [9], which complicates screening and leads to the need to analyze electrophysiological parameters, the clinical assessment of muscle fatigue [10].

Several studies have reported that in addition to systemic immune response, myasthenia gravis is also associated with mitochondrial dysfunction, leading to abnormal functioning of neuromuscular tissues, as well as affecting the proliferation and differentiation of immune cells [11]. Dysregulated mitochondrial functioning can lead to persistent oxidative stress, metabolic dysfunction, and autophagy [12].

It has been established that more than 30% of genes in the human genome are subject to post-transcriptional regulation of expression by microRNA [13]. The large number of microRNAs, among other epigenetic modifiers, can influence mitochondrial functions, exerting pro- or antioxidant effects, regulating oxidation-reduction reactions, and also controlling the bioenergetic status of the cell [14].

The available published data have reported altered microRNA expression profiles in patients with myasthenia gravis; for example, Lu et al. (2013) found an increase in hsa-miR-146a expression [15], while Liu et al. (2016) found an association between myasthenia gravis and hsa-miR-15a [16]. A detailed analysis of all myasthenia-associated microRNAs is presented in the review papers of Wang et al. (2020) [17] and Sabre et al. (2020) [18]; we will not dwell on them in detail. In most cases, microRNA is considered as a regulator of the immune response, as well as a regulator of processes involved in thymus function, while metabolic processes controlled by mitochondria are not considered.

In this study, the main object was microRNA potentially involved in the regulation of mitochondrial function in patients with myasthenia gravis compared with healthy volunteers. Additionally, we compared the plasma concentration of mitochondrial complex proteins. The aim of our study is to investigate the role of microRNAs in mitochondrial function in myasthenia gravis.

## 2. Materials and Methods

The study of mitochondrial dysfunction biomarkers was performed in two separate steps: analysis of plasma microRNA expression and analysis of plasma protein concentration in patients with myasthenia gravis compared with healthy volunteers. The main steps of the study design are presented in Figure 1.

### 2.1. Selecting MicroRNAs for Analysis

To select microRNAs for subsequent plasma expression analysis, differential expression analysis of the GSE85452 [19] dataset deposited in the NCBI (Gene Expression Omnibus National Center of Biotechnology Information) GEO was performed at the link https://www.ncbi.nlm.nih.gov/geo/ (accessed on 20 June 2025). GSE85452 contained RNA expression data obtained using the Illumina HumanHT-12 V4.0 expression beadchip microarrays, while peripheral blood monocytes from patients with myasthenia gravis (concordant and dixoncordant twins) were studied in comparison with healthy controls. Gene expression analysis was performed in the GSE85452 dataset obtained using the Illumina HumanHT-12 V4.0 expression beadchip platform. CD14 purified from PBMC served as samples.

In contrast to the original study, which divided participants into three groups: (1) singleton healthy controls, (2) discordant twins without clinical phenotype, and (3) MG patients including MG concordant and discordant twins and MG singleton patients. We formed the groups as follows: (1) general myasthenia group, including MG singleton patients and MG concordant and discordant twins; (2) control group, including singleton healthy controls and discordant twins without clinical phenotype. Additionally, we did not set fold change thresholds as in the original study (fold change > or <1.2); all genes whose log2 fold change did not equal zero with an FDR-corrected *p*-value < 0.05 were considered differentially expressed.

Microarray data were preprocessed using the lumi package (version 2.60.0) for R [20], and raw data were loess-normalized. Subsequent differential expression analysis was performed using the limma package (version 3.64.3) for R [21]. Data visualization was performed using the Glimma package (version 2.19.2) for R [22]. Genes with adjusted *p*-value < 0.05 were considered differentially expressed.

The obtained differentially expressed genes (DEGs) were then subjected to Gene Set Enrichment Analysis (GSEA) using the Illumina HumanHT-12 V4.0 gene set as the background genes. GSEA was performed using the ClusterProfiler package for R [23] and the WEB-based GEne SeT AnaLysis Toolkit (https://2024.webgestalt.org, accessed on 20 June 2025). The following terms were used: Biological Process, Molecular function, Cellular component, Reactome, KEGG. Clusters with FDR < 0.05 were considered significant.

After functional annotation, the most interesting gene clusters associated with mitochondrial functioning were identified. The search for predicted microRNAs associated with the genes of interest was performed using the following databases: MAMI, PicTar, TargetRank, TargetScan, miRcode, BCmicrO, Cupid, microrna.org, CoMeTa, Cupid with *p*-value < 0.05.

### 2.2. Ethics and Patient Recruitment

The study was conducted in accordance with the recommendations of the Declaration of Helsinki [24] and approved by the Local Ethics Committee of the Krasnoyarsk State Medical University named after Professor V.F. Voyno-Yasenetsky (extract from the protocol extract from protocol No. 116/2022 dated 27 December 2022). All patients and healthy volunteers signed informed consent to participate in the study and to publish the obtained data.

To study plasma microRNA expression, patients diagnosed with Myasthenia Gravis (*n* = 50) were consecutively recruited into the experimental group. Diagnoses were established by a specialist in the field of neurological disorders.

The inclusion criteria were as follows:Confirmed diagnosis of Myasthenia Gravis;Age 18–55 years;No signs of infectious disease during the collection of samples.

The control group (*n* = 31) included healthy volunteers of the appropriate age, gender, without diagnosed neurological diseases and signs of infectious diseases, as well as without chronic somatic pathologies in the decompensation stage.

Comparative metrics of patients in the control and experimental groups are presented in Table 1.

The experimental and control groups were unbalanced by gender. This imbalance is explained by the fact that myasthenia is more common in women; however, we consider this information as one of the limitations of our study.

Whole blood from the cubital vein was collected in 2 vacuum tubes with EDTA and then subjected to the following:Centrifugation for microRNA expression analysis to separate the plasma fraction according to standard protocols [25];Centrifugation for enzyme immunoassay according to reagent manufacturers’ protocols. The separated plasma was aliquoted and stored in a low-temperature freezer at −80 °C until further analysis.

We include small sample sizes as a limitation of this study, which is due to the low prevalence of the disease.

### 2.3. Analysis of Plasma MicroRNA Expression

Total RNA was isolated from blood plasma using the RIBO-sorb kit (K2-1-Et-100, Central Research Institute of Epidemiology of Rospotrebnadzor; Moscow, Russia) according to the manufacturer’s protocol. DNAase I (3911.2000, diaGene, Mytishchi, Russia) was used to remove contaminants in the form of DNA. Quantitative analysis was performed using Qubit 4 (Thermofisher scientific, Waltham, MA, USA) and the Equalbit RNA HS Assay Kit (EQ211-01, Vazyme, Nanjing, China).

The isolated RNA (1 μg) was reverse-transcribed using specific stem-loop primers and the HiScript II Reverse Transcriptase cDNA Synthesis Kit (R211-01, Vazyme, Nanjing, China) according to the manufacturer’s protocol.

Real-time PCR was performed on a LightCycler 480-II (Roche, Basel, Switzerland) using a 2.5-fold reaction mixture for RT-PCR in the presence of SYBR Green I dye (M 427, Syntol, Moscow, Russia), as well as synthesized Forward and Reverse primers. Primers were designed using sRNAprimerDB software (http://www.srnaprimerdb.com) and subsequently analyzed for dimerization using Multiple Primer Analyzer (https://www.thermofisher.com/ru/ru/home/brands/thermo-scientific/molecular-biology/molecular-biology-learning-center/molecular-biology-resource-library/thermo-scientific-web-tools/multiple-primer-analyzer.html, accessed on 10 June 2025).

MicroRNA expression was calculated using the Livak method [26]. Expression exceeding 40 cycles during fluorescence detection was excluded from further analysis. hsa-miR-191-3p was used as a reference microRNA due to its stable detection in blood plasma [27] and also due to the lack of association with differentially expressed genes in myasthenia.

### 2.4. Enzyme-Linked Immunosorbent Assay of Plasma

Two selected proteins were analyzed: Coenzyme Q10 (COQ10A), Glycerol-3-phosphate dehydrogenase (GPDH1), and Succinate dehydrogenase complex subunit C (SDHC) using ELISA Kit for Coenzyme Q10 Homolog A (COQ10A) (Cloud-Clone Corp, Katy, Katy, TX, USA), Human Glycerol-3-phosphate dehydrogenase (GAPDH1) ELISA Kit (BlueGene Biotech, Shanghai, China), and Human Mitochondrial Succinate Dehydrogenase Cytochrome B560 (SDCH) ELISA kit (BlueGene Biotech, Shanghai, China) according to the manufacturer’s protocol. Optical density and sample concentration were determined using a CLARIOstar Plus multimodal reader (BMG LABTECH, Ortenberg, Germany).

### 2.5. Analysis of the Data Obtained

The analysis of the distribution of microRNA expression and plasma protein concentration data was performed using the Shapiro–Wilk statistical test. The Median, 25th, and 75th percentiles (Me [LQ; UQ]) were used to describe the data with non-normal distribution. Outliers were detected using the Grubbs test.

For comparison of quantitative data, the nonparametric statistical Mann–Whitney test with Bonferroni correction and the nonparametric Kruskal–Wallis test with post hoc analysis based on the U-test were used. Statistical tests were implemented using the scipy.stats package for Python 3. Between-group differences were statistically significant at *p*-value < 0.05. ROC (Receiver Operator Curve) analysis was implemented using the scikit-learn package for Python 3.

To determine the protein concentration in the sample after ELISA, a 4-parameter logistic regression was constructed using Prism (https://www.graphpad.com/features, accessed on 15 May 2025).

## 3. Results

The GSE85452 dataset was used to identify differentially expressed genes, and the analysis was performed between the myasthenia (MG) and control groups. Multidimensional scaling (MDS) was performed on loess-normalized expression values. (Figure 2), with the largest variability described by the first two components. The normalized data then were subjected to differential expression analysis.

A total of 790 significantly differentially expressed genes were identified, with an adjusted *p*-value < 0.05. Among them, 333 were hypoexpressed and 457 were hyperexpressed (Figure 3).

The analyzed differentially expressed genes were subjected to Gene Set Enrichment analysis. All genes identified in the GSE85452 dataset were used as background genes.

The largest number of genes was enriched in non-specific Gene Ontology terms: a total of 424 fell into the metabolic process category, 353 into the biological regulation category, and 268 into the response to stimulus category. Nuclear genes (321) and membrane lumen genes (307) were also most frequently encountered (Figure 4).

The gene ontology term enrichment results were further post-processed to reduce redundancy using the Affinity propagation method. This resulted in a gene network that included terms such as regulation of immune system processes, autophagy, RNA splicing, and RIG-1 signaling (Figure 5). The maximum number of genes in a cluster was 2000, and the minimum was 5. Clusters with FDR < 0.05 were considered significant.

Also, using the ShinyGO 0.82 software (https://bioinformatics.sdstate.edu/go/, accessed on 25 May 2025), we obtained a separate cluster of genes involved in energy metabolism (Figure 6).

During the GSE analysis, we identified the clusters of most interest to us, including the energy metabolism clusters and the RIG-1 signaling cluster. The first four clusters by cellular components are most significantly enriched in mitochondria (FDR = 0.00002, enrichment ratio = 1.7647) and mitochondrial membranes (FDR = 0.00007, enrichment ratio = 2.1493). The genes of these clusters are presented in Table 2.

The RIG-I signaling pathway was further elucidated due to its ability to activate MAVS (mitochondrial antiviral-signaling protein) on mitochondria and peroxisomes [28] and, through a further molecular cascade, participate in glucose metabolism [29].

For the entire pool of genes of interest, a microRNA–Gene Targets interaction network was constructed. The analysis included microRNAs from databases containing predicted microRNA target genes with FDR < 0.05. The reduced network is shown in Figure 7. MicroRNAs interacting with a smaller proportion of target genes were trimmed.

To identify the relation of microRNAs associated with the above-mentioned gene targets with mitochondrial functions, miRNA Enrichment Analysis (miEAA, https://ccb-compute2.cs.uni-saarland.de/mieaa/, accessed on 25 May 2025) was performed for miRNAs associated. Among the obtained terms, the most interesting was GO:0005739 Mitochondrion (FDR adjusted *p*-value = 0.006), which included the following miRNAs: hsa-miR-27b-3p, hsa-miR-194-5p, hsa-miR-148a-3p, hsa-miR-766-3p, hsa-miR-181d-5p, hsa-miR-27a-3p, hsa-miR-181a-5p, hsa-miR-31-5p, hsa-miR-106a-5p, and GO:0070059 intrinsic apoptotic signaling pathway in response to endoplasmic reticulum stress enriched with hsa-miR-194-5p, hsa-miR-204-5p, hsa-miR-148a-3p, hsa-miR-590-3p, hsa-miR-766-3p, hsa-miR-181d-5p, hsa-miR-301b-3p, hsa-miR-181a-5p, hsa-miR-143-3p, hsa-miR-211-5p, hsa-miR-31-5p, and hsa-miR-106a-5p. The two presented terms had seven overlapping miRNAs: hsa-miR-148a-3p, hsa-miR-766-3p, hsa-miR-194-5p, hsa-miR-181d-5p, hsa-miR-181a-5p, hsa-miR-31-5p, hsa-miR-106a-5p. In the course of our work, we randomly selected three microRNAs: hsa-miR-194-5p, hsa-miR-181a-5p, hsa-miR-148a-3p. Their gene targets obtained as DEGs in microarray analysis are presented in Table 3.

Further, the microRNAs we selected were subjected to plasma expression analysis by RT-PCR with reverse transcription within two groups: patients with myasthenia gravis and healthy volunteers.

The normality test demonstrated that in most cases the expression data of the selected microRNAs did not follow the laws of normal distribution (*p*-value < 0.05).

After removing outliers using the Grubbs test, the myasthenia group included 49 participants, the control group 31 participants. The analysis revealed a statistically significant increase in hsa-miR-194-5p expression: Log10 Fold Change (L10FC) = 1.46, *p*-value < 0.0001, 95% CI [−6.3600; −3.2750], test power was 0.99, as well as a statistically significant decrease in hsa-miR-148a-3p expression: L10FC = −0.65, *p*-value = 0.02, 95% CI [0.36; 3.63], test power = 0.71; required sample sizes for group = 49 and 31. *p*-value was adjusted with Bonferroni correction.

In the case of hsa-miR-181a-5p, a slight decrease in expression was observed: L10FC = −0.08, *p*-value = 0.57, 95% CI [−0.8650; 1.3901]. The data described above are presented in Figure 8.

We also constructed ROC curves to assess the quality of the binary classification of study participants into patients with myasthenia and healthy volunteers. A quantitative interpretation of ROC is given by the AUC (Area Under Curve) indicator—the area under the curve, limited by the ROC curve and the axis of the proportion of false positive classifications. The higher the AUC indicator, the better the classifier, while a value of 0.5 demonstrates the unsuitability of the selected classification method.

Thus, in particular, for hsa-miR-148a-3p, AUC was equal to 0.66, for hsa-miR-194-5p AUC = 0.86, which characterizes the expression of this microRNA as a good biomarker for discrimination of myasthenia. For hsa-miR-181a-5p, AUC = 0.54, which demonstrates the unsuitability of its use in discrimination. All ROC curves are presented in Figure 9. Also, we additionally constructed a ROC curve for all microRNAs; its AUC = 0.92.

No statistically significant differences in microRNA expression depending on gender were found: hsa-miR-181a-5p: *p*-value = 0.72; hsa-miR-148a-3: *p*-value = 0.55 *p*; hsa-miR-194-5p: *p*-value = 0.58. Within the myasthenia gravis group, we also performed an analysis between genders; for microRNA hsa-miR-148a-3p, no statistically significant differences in expression were found (*p*-value = 0.54), as well as for hsa-miR-194-5p (*p*-value = 0.46), and no statistically significant difference was observed for hsa-miR-181a-5p (*p*-value = 0.89). Additionally, an analysis of the correlation of microRNA expression with age was performed using the Spearman method; the results of the analysis are shown in Table 4.

We performed an enzyme immunoassay of plasma to compare the concentration of proteins: COQ10A, GAPDH1, and SDHC in patients with myasthenia compared to controls. These proteins were randomly selected from a cluster of genes enriched for the following Gene Ontology terms: GO:0005739 Mitochondrion, GO:0045333 Cellular respiration.

The analysis revealed that the concentration of COQ10A in the plasma of patients with myasthenia was significantly lower than in healthy volunteers: 0.911 [0.439; 1.608] versus 1.815 [1.033; 2.916], respectively, *p*-value = 0.01, 95% CI [0.2433;1.2632], test power = 0.66, AUC = 0.67 (Figure 10). Within the myasthenia gravis group, we also performed an analysis between genders, and no statistically significant differences in concentration were found (*p*-value = 0.31).

For GPDH1, elevated plasma concentration was observed in patients with myasthenia: 16.144 [13.569; 19.979] vs. 14.967 [11.831; 19.600], respectively, but it was not classified as statistically significant: *p*-value = 0.23, 95% CI [−3.6233; 1.1210]. SDHC concentration was not determined because it was below detection limits. No correlation between protein concentration and age was found: COQ10A: Rho = 0.09, *p*-value = 0.43; GPDH1: Rho = 0.005, *p*-value = 0.97.

Thus, we found that myasthenia is characterized by aberrant expression of hsa-miR-148a-3p and hsa-miR-194-5p microRNAs, as well as reduced COQ10A concentrations.

## 4. Discussion

Our study revealed statistically significant differences in COQ10A (*p*-value = 0.01) concentrations and hsa-miR-148a-3p and hsa-miR-194-5p (*p*-value = 0.02 and <0.0001, respectively) expression levels in patients with myasthenia gravis compared to healthy volunteers.

Mitochondria play one of the key roles in maintaining cellular homeostasis. The slightest change in their functionality can induce a cascade of pathological events. Thus, it has been established that abnormal functioning of mitochondria is associated with excessive generation of reactive oxygen species (ROS) and subsequent oxidative stress [30]. Damage caused by ROS induces subsequent immunogenic processes that can lead to aggravation and persistence of the pathological process. Thus, autoimmune damage to mitochondria associated with the occurrence of a pro-oxidant state (excessive production of ROS) can be postulated in the pathogenesis of autoimmune diseases, in particular myasthenia gravis [31]. Thus, the role of mitochondrial dysfunction has been established for a number of autoimmune diseases [32].

An analysis of published data did not reveal any reports of association of hsa-miR-194-5p with myasthenia gravis. Xiao et al. (2022) established that hsa-miR-194-5p is reported to be involved in the regulation of apoptosis and is also associated with endoplasmic reticulum stress [33]; also, Nie et al. (2018) reported that increased expression of this microRNA in mice induces, among other things, mitochondrial dysfunction [34]. It was also reported that hsa-miR-194-5p is involved in the PI3K/AKT pathway, regulation of protein adhesion, MAPK pathway, regulation of cytokines and inflammatory response, and angiogenesis [35]. Some of the authors hypothesize a compensatory function of increased expression of this microRNA under conditions of oxidative stress and abnormal mitochondrial functioning.

There are also no data on the expression of hsa-miR-148a-3p in connection with the pathogenesis of myasthenia gravis. In the work of Anastasio et al. (2024), endothelial cells treated with interleukin 6 (IL-6) showed signs of oxidative stress and apoptotic death, along with a decrease in the expression of hsa-miR-148a-3p [36]; our study also demonstrated a decrease in plasma expression of this microRNA. Also, the authors established an association of hsa-miR-148a-3p and the SIRT7 gene and put forward a hypothesis about the role of this microRNA in counteracting SIRT7-induced apoptosis and mitochondrial damage.

Reduced expression pattern of hsa-miR-181 in PBMC (Peripheral blood mononuclear cell) of patients with myasthenia gravis was reported in the work of Zhang et al. (2016) [37]. In the same work a correlation was established between the expression pattern of this microRNA and plasma levels of proinflammatory cytokines such as IL-7 and IL-17. Our results are consistent with these data: hsa-miR-181a-5p expression was lower in patients with myasthenia gravis compared to healthy volunteers, but this difference was insignificant, which may be due to the limitations of our sample; further research in this direction may confirm or refute the existing data. miR-181 subtypes immune function in thymic epithelial cells is established, as well as the importance of this microRNA family in the control of thymocyte development and mature T cell export [38]. An association between miR-181a and transforming growth factor receptor β (Tgfbr1) is also demonstrated [39].

Coenzyme Q10A (COQ10A) or ubiquinone is part of the electron transport chain and plays a key role in energy and redox balance [40]. In the work of C. Andreani et al. (2018), treatment of presarcopenic mice with ubiquinol supplementation and physical exercise improved the overall condition of skeletal muscles by inhibiting caspase-induced apoptosis and autophagy, which demonstrates the important role of ubiquinone in maintaining muscle function [41].

Interestingly, although miRNA 194-5p does not directly target ubiquinone, according to the MirTarBase database, its validated targets are COQ6 and COQ9 (experiment: HITS-CLIP), enzymes involved in ubiquinone biosynthesis, in particular through modification of the quinone head group via deamination [42]. Moreover, a weak but statistically significant negative correlation (Spearman correlation: Rho = −0.27, *p*-value = 0.017) was found between the fold change expression of this microRNA and plasma ubiquinone concentrations. According to this fact, we can assume an indirect association between the expression of this microRNA and the concentration of ubiquinone.

In addition, also in accordance with the Tarbase, the hsa-miR-181a-5p microRNA (with reduced but not statistically significant expression in our study) directly validated targets Glycerol-3-Phosphate Dehydrogenase 1 (HITS-CLIP). The observed tendency towards decreased expression of this microRNA and increased protein concentration may have a strong association that requires further confirmation.

An abnormal expression of the microRNAs and changed protein concentration we studied may be both a consequence and one of the links in the pathological process occurring in myasthenia gravis. Dysregulated expression of these microRNAs and aberrant plasma protein concentration may be an indicator of mitochondrial dysfunction accompanying and inducing the inflammatory process.

Mitochondrial dysfunction entails disruption of normal cellular metabolism and, as a consequence, leads to an increase in the production of reactive oxygen species and triggers a cascade of molecular events leading to the persistence of inflammation, and therefore the preservation and progression of the pathological process. Therefore, one of the important strategies is the control of the quality of mitochondrial function; moreover, therapeutic approaches aimed at restoring and improving mitochondrial function seem promising [11]. It is important to note that long-term use of immunosuppressants, which are one of the main approaches to treating myasthenia gravis, can also negatively affect mitochondrial function [43]. Search and validation of biological markers of mitochondrial dysfunction in patients with myasthenia gravis and other neuromuscular diseases, and their validation and implementation in clinical practice can be considered one of the priority areas. Additionally, markers of mitochondrial dysfunction can be used as indicators of the necessity to change the therapeutic approach.

As limitations of this study, we note the lack of a validation experiment on an independent sample and small sample sizes.

## 5. Conclusions

We conducted a study of plasma expression of microRNA hsa-miR-181a-5p, hsa-miR-194-5p, and hsa-miR-148a-3p, as well as three proteins—COQ10A, GAPDH1, and SDHC—in patients with an established diagnosis of myasthenia gravis in comparison with healthy volunteers. The analysis suggests a relationship between these microRNAs and proteins with mitochondrial dysfunction accompanying the pathological process. Abnormal expression of two microRNAs hsa-miR-194-5p and hsa-miR-148a-3p in the plasma of patients with myasthenia, as well as a decrease in the concentration of COQ10A, was established. The obtained data allow these molecules to be used as a potential additional diagnostic tool for mitochondrial dysfunction.

## Figures and Tables

**Figure 1 medsci-13-00118-f001:**
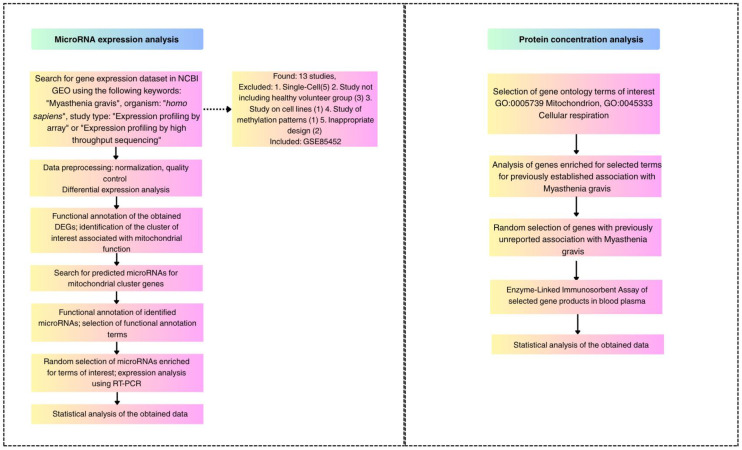
Flow Chart representing study design step by step.

**Figure 2 medsci-13-00118-f002:**
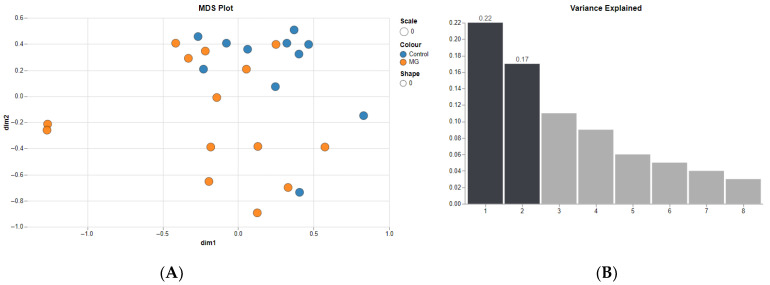
(**A**) Multidimensional scaling clustering of samples. (**B**) Bar plot to visualize the amount of variance explained by each component.

**Figure 3 medsci-13-00118-f003:**
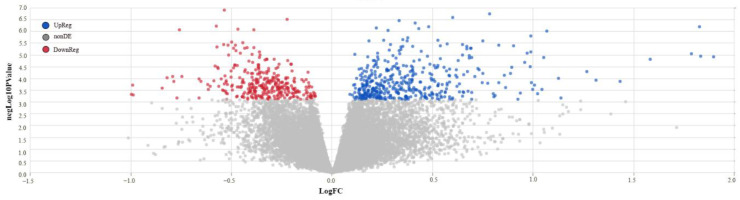
Differentially expressed genes.

**Figure 4 medsci-13-00118-f004:**
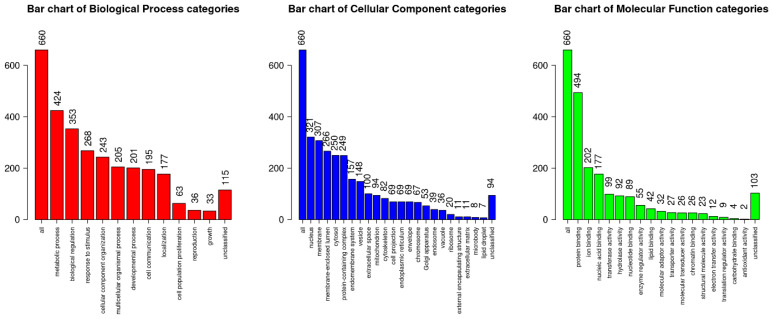
Enrichment of differentially expressed genes by gene ontology terms.

**Figure 5 medsci-13-00118-f005:**
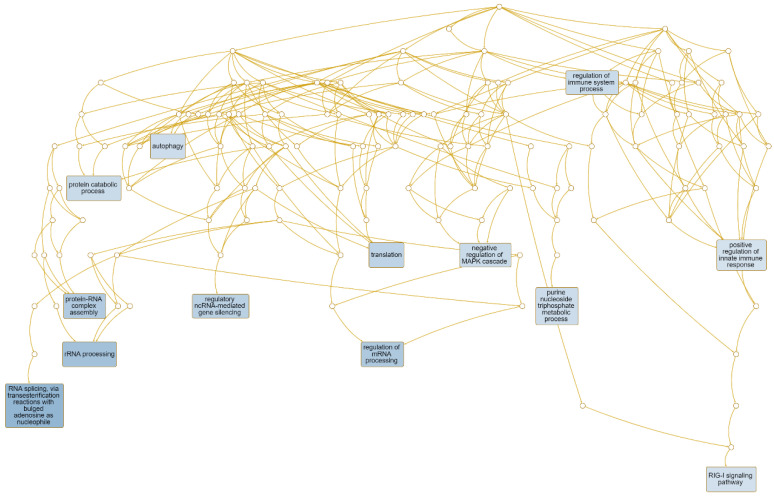
Network and Volcano plot Gene Ontology terms reduced by affinity propagation: network.

**Figure 6 medsci-13-00118-f006:**
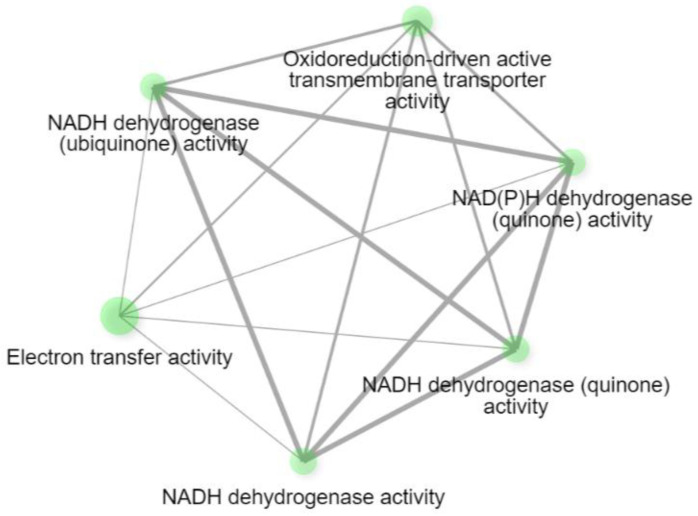
Gene network enriched for gene ontology terms associated with cellular energy metabolism.

**Figure 7 medsci-13-00118-f007:**
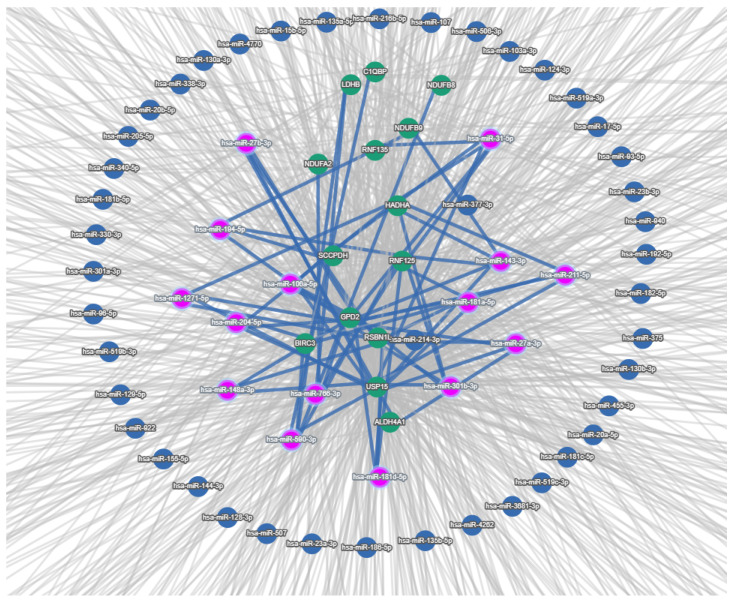
Network of microRNAs and predicted gene targets.

**Figure 8 medsci-13-00118-f008:**
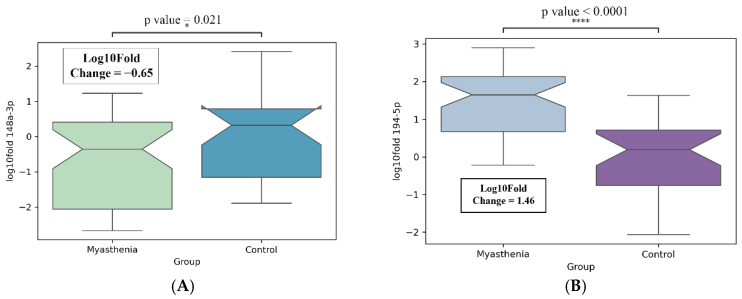
MicroRNA expression in patients with myasthenia (*n* = 49) compared to controls (*n* = 31). (**A**) hsa-miR-148a-3p; (**B**) hsa-miR-194-5p; (**C**) hsa-miR-181a-5p. * *p*-value < 0.05; **** *p*-value < 0.0001; ns-non significant.

**Figure 9 medsci-13-00118-f009:**
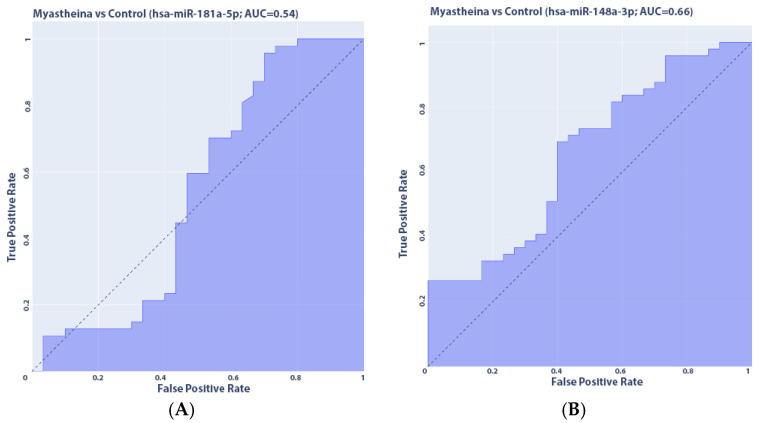
ROC (receiver operating characteristic) curves for the studied microRNAs: (**A**) hsa-miR-181a-5p; (**B**) hsa-miR-148a-3p; (**C**) hsa-miR-194-5p; (**D**) all microRNAs.

**Figure 10 medsci-13-00118-f010:**
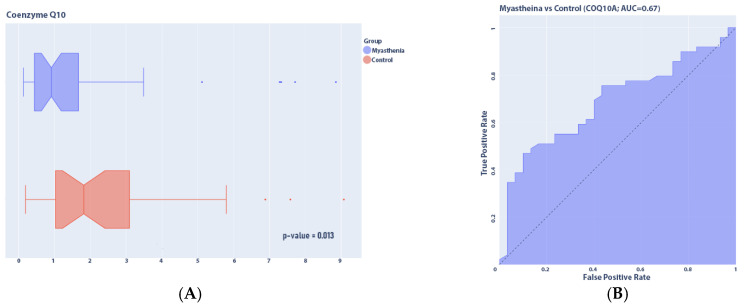
COQ10A concentration in plasma of patients with myasthenia (*n* = 49) compared to controls (*n* = 31). (**A**) Distribution of concentrations; (**B**) ROC (receiver operating characteristic) curve.

**Table 1 medsci-13-00118-t001:** Clinical characteristics of patients included in the study.

Characteristic	Patients with Myasthenia Gravis	Healthy Volunteers	Significance Level (*p*-Value)
Gender distribution			
Males	19.61%	46.67%	0.01 (X^2^)
Females	80.39%	53.33%	
Age, Me [Q1; Q2]	37.00 [28.50; 41.50]	35.00 [27.00; 40.75]	0.40 (U test)

**Table 2 medsci-13-00118-t002:** Clusters of GO terms and genes included in them.

GO	Genes and its Expression	FOLD	FDR
GO:0016491: oxidoreductase activity	HADHA ↓, OGDH ↑, IMPDH1 ↓, LDHB ↓, STEAP3 ↑, FDX1 ↓, ALDH4A1 ↓, MSRA ↓, LTC4S ↓, MICAL2 ↑, DECR2 ↓, CIAPIN1 ↓, COX6A1 ↓, GPD2 ↑, SDHB ↓, PRDX4 ↓, ALKBH8 ↑, SCCPDH ↓, RSBN1L ↑, NDUFS3 ↓, DUS4L-BCAP29 ↑, NCF1 ↑, NDUFA2 ↓, NDUFB10 ↓, NDUFB9 ↓, NDUFB8 ↓, NDUFA4 ↓, COX5A ↓	4.2	0.03
GO:0009055: electron transfer activity	FDX1 ↓, CIAPIN1 ↓, COX6A1 ↓, SDHB ↓, NDUFS3 ↓, NDUFA2 ↓, NDUFB10 ↓, NDUFB9 ↓, NDUFB8 ↓, NDUFA4 ↓, NCF1 ↑, ALDH4A1 ↓, COX5A ↓	3.7	0.01
GO:0008137: NADH dehydrogenase (ubiquinone) activity	MT-ND4L ↓, MT-ND1 ↓, NDUFS1 ↑	5.3	0.03
GO:0003954: NADH dehydrogenase activity	NDUFB10 ↓, NDUFA8 ↑, NDUFS1 ↑, NDUFA5 ↓, NDUFS8 ↓, NQO1 ↑, NDUFB4 ↓, NDUFV1 ↑, NDUFA10 ↓, NDUFS4 ↓	4.9	0.04
GO:0039529: RIG-I signaling pathway	RNF135 ↑, RNF125 ↑, BIRC3 ↑, USP15 ↑, C1QBP ↓, NOP53 ↓	6.76	0.03

↓—downregulated; ↑—upregulated.

**Table 3 medsci-13-00118-t003:** microRNAs and gene targets.

MicroRNA	Gene Target	Predicted by
Hsa-miR-194-5p	RSBN1L RSBN1L RSBN1L SCCPDH RSBN1L NDUFB9 RSBN1L RNF125 RSBN1L RSBN1L	MAMI PicTar TargetRank TargetRank TargetScan TargetScan miRcode BCmicrO BCmicrO Cupid
Hsa-miR-181a-5p	RSBN1L GPD2 USP15 RNF125 GPD2 USP15 GPD2 USP15	MIRT707909 ElMMo3 MAMI TargetScan TargetScan TargetScan BCmicrO CoMeTa
Hsa-miR-148a-3p	RSBN1L RSBN1L RSBN1L RSBN1L RSBN1L RSBN1L USP15 GPD2	ElMMo3 MAMI PicTar TargetRank TargetScan microrna.org CoMeTa Cupid

**Table 4 medsci-13-00118-t004:** Analysis of the correlation of microRNA expression with the age of patients.

MicroRNA	Correlation Level (*r_s_*)	Significance Level (*p*-Value)
hsa-miR-148a-3p	−0.03	0.79
hsa-miR-181a-5p	0.11	0.36
hsa-miR-194-5p	−0.05	0.68

## Data Availability

The data presented in this study are available on request from the corresponding author. The data are not publicly available due to internal regulations.

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
