# Peer review of "Plasma Biomarkers of Mitochondrial Dysfunction in Patients with Myasthenia Gravis"

_medsci, 2025, doi:10.3390/medsci13030118_

Round 1
Reviewer 1 Report
Comments and Suggestions for Authors
The authors analyzed gene expression, miRNA expression, and plasma protein levels using both a publicly available dataset and prospectivly collected samples from a clinical study of Myasthenia Gravis-patients and controls. All molecular analyses are technically sound, and the statistical approaches appear appropriate.
Nevertheless, I have several concerns regarding the selection strategies employed across the different data layers. In particular, the criteria for selecting specific miRNAs, and proteins for follow-up appear inconsistent or insufficiently explained. The lack of a clear and systematic rationale for these choices is, in my view, the most problematic aspect of the study and should be addressed to improve transparency and reproducibility.
Abstract: RT-PCR RT shlud be written as RT-PCR
Introduction: "In this study, the main object of the study" should be written as "In this study, the main object was"
Material & Methods: Section 2.1: The first sentence should be revised. Unclear which background was used for GSEA. The "various databases" for miRNA search should be provided.
Line 177: Instead "identified" another verb like "measured" or "analysed" should be used.
The dataset used for differential expression analysis has been previously analyzed in at several (>3) other publications. While re-analysis can yield novel insights, the authors should acknowledge this prior work, compare key findings, and explain any discrepancies. This would improve transparency and help readers assess the novelty and robustness of the current analysis.
Figure 6: What are the grey lines? They seem to be connected to edges outside the plot?
The authors report 40-50 associated miRNAs, but follow up only on 3. Without a clear rationale for this selection, the approach appears somewhat arbitrary. A more systematic strategy would be preferable — ideally, functional analysis of all identified miRNAs. If that is not feasible, the authors should at least justify their selection criteria (e.g., based on effect size, biological relevance, or literature support) and discuss the remaining candidates more comprehensively.
Fig. 7: The numer of patients analysed should be provided in the caption or figures.
Lastly, the authors report measuring three genes in plasma samples, but it is unclear how these genes were selected. Were they chosen based on the differential expression analysis, predicted miRNA binding sites, or some other rationale? This connection needs to be clarified and also discussed(!), as it is critical for understanding the logic behind the experimental design and the interpretation of the findings.
Validation experiments in an independet cohort is missing and should be stated as limitation.
Author Response
Dear Reviewer! Thank you for your time and comments. We will try to clarify all ambiguous points step by step. All modifications made to the manuscript are highlighted in yellow.
- RT-PCR RT was replaced with RT-PCR in the abstract, thanks for noticing this typo.
- The incorrectly composed sentence "In this study, the main object of the study" in the introduction has been replaced.
- In the materials and methods, we clarified the issue with background genes. In enrichment analysis, you start with a list of genes of interest (your "input list"). This list is typically generated from a high-throughput experiment, like RNA sequencing or microarray analysis. The background gene set provides the context for your input list. It represents all the genes that were potentially measurable in your experiment
- The verb measured has been replaced with analyzed.
- In the Materials and Methods section, we have added a citation of the original study and we have clarified what exactly we changed in our approach.
- Yes, as it was written in the results, the picture shows only a part of the microRNA-gene target network, it was cut off by those microRNAs that were associated with most of the genes of the cluster we are interested in. There are more microRNAs outside the picture, the picture was cut off so that it was readable. If we left everything as it is in the original, it would be impossible to see the names of the microRNAs and genes.
- We added information about the selection of microRNAs for subsequent analysis by RT-PCR. Additional selection criteria were enrichment by functional annotation terms. This is a really important issue. And in our approach, there were seven microRNAs that fit the established criteria. We selected three of them, not all, due to the limited budget for the study, unfortunately.
- We included information about the number of patients in the figure captions.
- Indeed, questions may arise due to the insufficiently transparent design of the study. We have added a diagram to the materials and methods that will clarify these points. The goal was to identify plasma markers of myasthenia at the microRNA and protein levels. The fact is that microRNA expression studies and protein concentration studies are two different studies that are not related to each other. Despite this, a potential relationship was found between microRNA expression and plasma protein concentration - we have added information about this to the discussion.
- Information about the limitations of the study was also included in the discussions.
We hope we have clarified all the controversial points, best wishes, authors.
Reviewer 2 Report
Comments and Suggestions for Authors
My major comments are:
ABSTRACT:
- Authors must detail the sample number and the number of subjects per group.
- Re-write: “characterized by pathological fatigue of striated muscles” — correct, but could be phrased more naturally as “…characterized by fatigue of striated muscles due to impaired neuromuscular transmission.”
- Phrases such as “makes a significant contribution to the formation of metabolic abnormalities” are awkward. Consider “contributes significantly to metabolic abnormalities…”
- The results section is too concise and lacks quantitative data.
Instead of just saying: “A statistically increased expression of hsa-miR-194-5p and a statistically significant decrease in the expression of hsa-miR-148a-3p were found…” you should report actual fold changes, p-values, and confidence intervals.
Similarly, the reported decrease in COQ10A should be quantified.
- The conclusion should be better defined. It is not clear.
INTRODICTION
This paragraph outlines the objective and aim of the study clearly, but it has some redundancies, awkward phrasing, and minor scientific weaknesses. The authors should rewrite it
MATERIALS AND METHODS
Informed consent is not an inclusion criterion. It is a requirement of the research process.
RESULTS
First paragraph: In the text of the results section, authors should focus on presenting their results and not explaining the concept of the method that should go in the previous section.
DISCUSSION
In the first paragraph of the discussion, the authors must express the level of significance (p value) and should explain the reason for this finding.
Line 276, the reference is misspelled
Authors should include a section on practical application and limitations of the study within the discussion. They should be included as subsections.
CONCLUSION
The conclusion should be 2-3 sentences explaining the findings of this study. It should be concise and clear.
The style of reference is correct
Author Response
Dear Reviewer, Thank you for your time and important comments. We will try to answer all your questions and comments. All changes made to the manuscript are highlighted in yellow.
- We have indicated the number of participants in each group.
- We have rewritten the phrase “characterized by pathological fatigue of striated muscles” to be more stylistically correct.
- We have made changes to the style of the sentences you pointed out.
- We have added specific data on significance level and change factor in the results section.
- We tried to rewrite the conclusion.
INTRODICTION
- We have tried to make changes to the style of the introductory sentences, thank you for your comment.
MATERIALS AND METHODS
Thank you, we have removed the presence of informed consent from the inclusion criteria.
RESULTS
We have tried to make changes to the first paragraph of the results section.
DISCUSSION
- We have added all the quantitative data obtained in our study to the discussion.
- Thank you for pointing out the problems with the link, we have modified it.
- We included information about the study's limitations and potential practical implications in the discussion.
CONCLUSION
We have tried to modify and shorten the conclusion section.
We hope we have clarified all the controversial points, best wishes, authors.
Reviewer 3 Report
Comments and Suggestions for Authors
The manuscript " Plasma Biomarkers of Mitochondrial Dysfunction in Patients with Myasthenia Gravis’’ compares the plasma biomarkers associated with mitochondrial dysfunction, such as, microRNAs and concentration of mitochondrial complex proteins in myasthenia gravis (MG) and healthy controls. The study addresses an under-explored but relevant area that aims to link mitochondrial defects with MG via plasma biomarkers that would contribute towards the diagnostic approach of MG.
After going through the manuscript, I have following comments for the authors:
- The methodology is abundantly described. However, the rationale behind choosing the microRNAs and concentration of mitochondrial complex proteins as biomarkers for MG is not clear. There are other common mitochondrial plasma biomarkers such as lactate, FGF21, GDF15. Why were these biomarkers not considered?
- Sample size and power calculation are missing. Was the sample size sufficient enough and the study adequately powered to justify the statistical outcomes?
- Sex- and age-related differences are frequently reported in biomarker levels in mitochondrial and autoimmune disorders. Were such differences reported in MG in this study?
The language is mostly fine. Minor grammatical corrections and syntax adjustments suggested.
Author Response
Dear Reviewer! Thank you very much for your time. We greatly appreciate your comments and feedback. We will try to answer all your questions. All modifications made to the manuscript are highlighted in yellow.
- We added an additional figure in the Materials and Methods section to clarify the design. 7. We added information about the selection of miRNAs for subsequent RT-PCR analysis. Additional selection criteria were enrichment for functional annotation terms. This is a really important issue. And in our approach, there were seven miRNAs that met the established criteria. Unfortunately, we selected three of them, not all, due to the limited budget for the study. Regarding proteins, we also added information about the selection of proteins for analysis. An important point that was not clear before our revision is that the study of miRNA expression and the measurement of protein concentration are two parallel processes that were not linked by a single biological logic, despite the fact that their relationship was suggested by us in the discussion section.
As for lactate, FGF21, GDF15 - these molecules have already been studied in connection with their association with the pathogenesis of myasthenia, so we did not include them in our work.
- Information about sample size and power calculation of the study has been added to the results, thank you for your comment.
- Some of the information about gender and age relationships was in the original version of the manuscript, but we also added additional information.
We hope we have clarified all the controversial points, best wishes, authors.
Round 2
Reviewer 1 Report
Comments and Suggestions for Authors
The authors have responded successfully to my previous comments. I now have only a minor issues remaining. Specifically, the figure captions are currently too brief. The authors should expand on the captions to provide more detailed explanations, and they should avoid using abbreviations without definitions.
Author Response
Dear Reviewer, Thank you for your comments and observations.
We have tried to describe each of the figures presented in the manuscript in more detail.
Best wishes, the authors.
Reviewer 2 Report
Comments and Suggestions for Authors
The authors have made a great effort to improve the article and have responded to all my comments satisfactorily.
Author Response
Dear Reviewer, Thank you for your comments and evaluation of our work.
Best wishes, authors.